# Covalent DNA Binding Is Essential for Gram-Negative Antibacterial Activity of Broad Spectrum Pyrrolobenzodiazepines

**DOI:** 10.3390/antibiotics11121770

**Published:** 2022-12-07

**Authors:** Pietro Picconi, Charlotte K. Hind, J. Mark Sutton, Khondaker Miraz Rahman

**Affiliations:** 1Institute of Pharmaceutical Science, King’s College London, London SE1 9NH, UK; 2UK Health Security Agency, Research and Evaluation, Porton Down, Salisbury SP4 0JG, UK

**Keywords:** antimicrobial resistance, broad-spectrum antibiotics, antibacterial drug discovery, gram-negative bacteria, pyrrolobenzodiazepines

## Abstract

It is urgent to find new antibiotic classes against multidrug-resistant bacteria as the rate of discovery of new classes of antibiotics has been very slow in the last 50 years. Recently, pyrrolobenzodiazepines (PBDs) with a C8-linked aliphatic-heterocycle have been identified as a new broad-spectrum antibiotic class with activity against Gram-negative bacteria. The active imine moiety of the reported lead pyrrolobenzodiazepine compounds was replaced with amide to obtain the non-DNA binding and noncytotoxic dilactam analogues to understand the structure-activity relationship further and improve the safety potential of this class. The synthesised compounds were tested against panels of multidrug-resistant Gram-positive and Gram-negative bacteria, including WHO priority pathogens. Minimum inhibitory concentrations for the dilactam analogues ranged from 4 to 32 mg/L for MDR Gram-positive bacteria, compared to 0.03 to 2 mg/L for the corresponding imine analogues. At the same time, they were found to be inactive against MDR Gram-negative bacteria, with a MIC > 32 mg/L, compared to a MIC of 0.5 to 32 mg/L for imine analogues. A molecular modelling study suggests that the lack of imine functionality also affects the interaction of PBDs with DNA gyrase. This study suggests that the presence of N10-C11 imine moiety is crucial for the broad-spectrum activity of pyrrolobenzodiazepines.

## 1. Introduction

Antimicrobial resistance (AMR) represents a significant challenge to future healthcare provision [1]. Developing novel antimicrobial agents for treating infections caused by multidrug-resistant (MDR) Gram-negative pathogens is an urgent priority [2,3]. However, the discovery of new antibiotics effective against Gram-negative bacteria is a major challenge. Moreover, the relatively low commercial return on investment for new antimicrobials has resulted in a lack of willingness from pharmaceutical companies to initiate antibiotics discovery programmes [4,5]. The scientific community and the industry have not been successful in discovering new chemical scaffolds with antimicrobial activity, and the research has focused on existing chemical classes [6]. This has led to a discovery void, and no new classes of antibiotics with activity against Gram-negative bacteria have been marketed since the discovery of daptomycin [4] in the late 1980s. Therefore, it is important to explore diverse chemical scaffolds to develop new antibiotics that can be used to treat multiple-drug-resistant infections. Recently, C8-linked pyrrolobenzodiazepines with a third aliphatic-heterocycle have been identified as a new broad-spectrum antibiotic class with activity against MDR bacteria, including WHO priority pathogens [7]. This provided an important new chemical scaffold in our search for new broad-spectrum antibiotics and replenished the waning pipeline of antibiotic discovery. Therefore, it is essential to explore the structure-activity relationship of this new chemical class, to assess the scope of chemical modifications and reduce the overall toxicity typically associated with PBDs.

Natural PBDs are covalent DNA minor groove binding molecules produced by *Streptomyces* bacteria [8,9] and other species. PBDs have a soft N10-C11 imine electrophile that can covalently bind to guanine bases [10] to form either monoalkylated adducts or cross-linked adducts in the case of PBD dimers [11]. They have been extensively studied as anticancer agents [12,13,14,15,16,17,18], and a large number of PBDs are being clinically evaluated as payloads for antibody-drug conjugates (ADCs), demonstrating the broad therapeutic utility of this chemical class [19,20,21,22,23]. More recently, PBDs have been explored for their antibacterial activity. While PBD monomers have shown activities against both Gram-positive and Gram-negative bacteria [24,25], PBD dimers are only active against Gram-positive bacteria due to their inability to cross Gram-negative membranes [26]. PBD monomers have a better toxicity profile compared to PBD dimers [27] and have shown some promise as a new antibacterial class [7,28]. We recently described a series of C8-linked PBD monomers with a third aliphatic heterocycle with broad-spectrum activity against ESKAPE pathogens. To further explore the structure-activity relationship and to evaluate the importance of the N10-C11 imine moiety in conferring antibacterial activity, we synthesised dilactam analogues of compounds 7 and 8 (Figure 1), which were identified as lead compounds with excellent activity against both Gram-positive and Gram-negative bacteria [7]. PBD dilactams cannot interact covalently with DNA due to the absence of the reactive imine group and hence do not show cytotoxicity normally associated with PBD chemical class [29]. The synthesised dilactam analogues were found to be moderately active against Gram-positive bacteria, while there was a complete loss of activity against Gram-negative bacteria. While the study confirms the importance of the reactive imine group and possibly covalent DNA binding for the broad-spectrum activity, it offers new opportunities to obtain nontoxic new antibacterial compounds against Gram-positive bacteria. This is the first report of an antimicrobial evaluation of C8-linked PBD dilactam monomers. 

## 2. Results and Discussion

The synthesis of dilactam analogues of compounds 7 and 8 was based on a convergent synthetic strategy, which contemplates the formation of the dilactam-modified PBD core and the C8 correspondent tail, to then be coupled to give the final desired product. The process for synthesising the dilactam-modified PBD core is reported in Figure 1. The synthetic route is based on previously reported literature for this class of compounds [29] and shares several similarities with the synthetic route used for the traditional PBD core, with key reactions involved in forming the two amide bonds of the dilactam ring. 

The synthetic route started from the carboxylic acid intermediate 1a that underwent amide coupling with (*S*)-proline methyl ester, using acyl chloride activation strategy, giving intermediate 1b (Figure 1) in reasonable yield (36%) after purification by column chromatography. Successive metal catalysed reduction in the nitro group of the derivative 1b brought directly to the intramolecular closure of the dilactam ring, giving compound 1c. The reductive hydrogenation catalysed by Raney-Ni under the pressure of H_2_ converted the nitro group into the correspondent aniline that promoted the favoured intramolecular nucleophilic attachment to the C11 atom, causing the one-pot formation of the amide bond after eliminating methanol. In some cases, the spontaneous formation of the amide bond occurred partially after hydrogenation. When this occurred, the total conversion of 1b to 1c was obtained, heating the mixture of two compounds (after filtration of the catalyst) until the disappearance of the aniline intermediate. Intermediate 1c subsequently underwent aqueous basic hydrolysis to give the final dilactam-modified PBD core 1d with the total conversion of the starting material. 

For synthesising the final dilactam analogues of compounds 7 and 8, intermediate 1d was coupled to previously synthesised C8 tails, 2a and 2b (Figure 2 and ESI). EDCI and DMAP were used to activate the carboxylic acid, giving the formation of the final dilactam analogues 7a and 8a in reasonable yield (43–46%) after purification by column chromatography. In the case of dilactam derivatives, no further synthetic steps have been required after the amide coupling between core and C8 tails. This is due to the stability of the N10-C11 amide bond, which does not require any protective synthetic strategy to avoid its degradation, unlike N10-C11 imine-containing PBD synthesis, where Alloc-THP protection is required. The intermediate synthesis of the core and the final products were fully characterised by LC-MS, HRMS, and ^1^H and ^13^C NMR analysis (ESI). Compounds 7 and 8 were synthesised according to a procedure we recently described [7].

The synthesised dilactam derivatives 7a and 8a were screened for their antimicrobial activity against a panel of MDR Gram-positive and Gram-negative bacteria that contained strains representing a range of antibiotic resistance profiles. The results are reported, along with the MIC values for the standard PBD parent compounds, in Table 1 and Table 2. 

Compound 7a, a dilactam analogue of benzofuran-thiomorpholine PBD derivative 7, showed moderate activity against Gram-positive strains, with MIC values between 4 and more than 32 µg/mL, compared to values between 0.03 and 2 µg/mL for the parent compound. The same trend was observed for dimethyl benzothiophene dilactam derivative 8a, with MIC values between 8 and more than 32 µg/mL against Gram-positive strains, compared to the MIC values between 0.03 and 0.06 for the parent compound. However, the level of activity reported against the Gram-positive strains suggests that noncovalent interactions have a role in contributing to the antibacterial properties of the series. 

The dilactam analogues did not show any activity against the MDR Gram-negative bacteria tested with MIC values > 32 µg/mL. The results suggest that the lack of the imine moiety caused a complete loss of the antibacterial activity of the dilactam analogues, indicating how the presence of covalent interaction between the PBD derivatives and the DNA minor groove was fundamental for the antibacterial activity of the series.

The synthesised dilactam derivatives 7a and 8a were evaluated for their toxicity against the eukaryotic Wi-38 cell line using the MTT cytotoxicity assay. As expected, both 7a and 8a were less toxic compared to their imine counterparts, suggesting the role of covalent DNA binding in eukaryotic toxicity by PBD compounds (Figure 2). 

The C8-linked PBDs, including the recently reported broad spectrum series, appeared to work by a mixed mechanism of action, with covalent DNA interaction and DNA gyrase inhibition contributing to the overall antibacterial activity of the compounds [7,24]. It is well documented that the dilactam analogues of PBDs cannot covalently interact with DNA due to the lack of N10-C11 imine moiety. We investigated the influence of the imine group on the DNA gyrase interaction of these molecules using a molecular modelling study. We used two previously reported DNA sequences (5′-TAT-AGA-TAT-AGA-TAT-3′ and 5′-CGC-TAT-AGA-TAT-CGC-3′), and the DNA gyrase A from *Staphylococcus aureus* to study the interaction of dilactam 7a and the corresponding imine 7 with both DNA sequences and DNA gyrase enzyme. The results suggest that the dilactam 7a fits snugly within the DNA minor groove (Figure 3A and Appendix A). However, it occupies a different binding pocket within the DNA gyrase A compared to the active imine compound 7 (Figure 3B). Notably, there appears to be a different orientation with respect to the critical amino acid Ser85, which is important for anchoring fluoroquinolones within the binding groove. Therefore, the lack of covalent DNA attachment, which is associated with DNA strand stabilisation, and the altered binding to DNA gyrase, may both contribute to the reduction in antibacterial activity of PBD dilactams. The ability to noncovalently interact with the DNA and snugly fit within the DNA minor groove may explain why these dilactams retain some antibacterial activity in the Gram-positive bacteria.

## 3. Materials and Methods

### 3.1. General Chemistry

The solvent and the reagents utilised for synthesising the compounds were obtained from Sigma-Aldrich, Fluorochem, Alfa Aesar and Fisher Scientific. Plates coated with silica gel (from Merck, F254 plates, silica gel 60) were used to analyse reactions by thin layer chromatography (TLC). The TLC plates were visualised under ultra-violet radiation (UV) at either 254 nm or 365 nm to monitor the progress of the reactions. Liquid column chromatography and flash chromatography were used to purify the compounds using silica gel as a stationary phase (silica gel from Merck, 230–400 mesh). An RC-600 evaporating system (from KNF), equipped with an SC-920 G vacuum pump (KNF), was used to evaporate the solvents. Wet compounds in some reaction steps were dried using a VacuumTherm (from Thermo Scientific, Waltham, MA, USA) vacuum oven. A UN55 (Memmert, Büchenbach, Germany) oven was used to dry glassware at 200 °C before anhydrous reactions. A Spectrospin 400 MHz spectrometer (Bruker, Yokohama, Japan) equipped with an automated SampleXpress (Bruker) sample submission system was used to record the ^1^H and ^13^C nuclei nuclear magnetic (NMR) spectra (Appendix A). The chemical shifts were reported relative to trimethylsilane (TMS) used as the standard (0.00 ppm). Signals were identified and described as singlet (s), doublet (d), t (triplet), q (quartet), or m (multiplets). LC-MS analyses were performed on a Waters Alliance 2695 system (from Waters), with elution in a gradient using a previously reported method (Appendix A) [30]. HPLC grade solvents were used as the mobile phase, while a Monolithic C18 50 x × 4.60 mm column (from Phenomenex, Torrance, CA, USA) was used as the stationary phase. UV detection was performed using a Waters 2996 photo array detector (from Waters, Milford, MA, USA). Mass Spectra were collected in both ESI+ or ESI− modes by a Waters QZ instrument (from Waters) coupled to the HPLC system. HRMS analyses were performed on an Exactive HCD Orbitrap mass spectrometer (from Thermo Scientific), with spectra collected in either ESI+ and ESI- modes depending on the substrate. As previously described, the hydrogenation reactions were conducted using a pressurised hydrogenation apparatus (from Parr, New York, NY, USA) [31].

### 3.2. Synthetic Procedures and Characterisation of Compounds

Synthesis of methyl (5-methoxy-4-(4-methoxy-4-oxobutoxy)-2-nitrobenzoyl)-L-prolinate (1b):

Compound 1a (2 gm, 6.12 mmol, 1 equiv.), obtained according to a literature procedure [12], was dissolved in dry DCM and transferred to a round bottom flask. Oxalyl chloride (2.33 gm, 18.36 mmol, 3 equiv.) was added to the solution, followed by a catalytic amount of dry DMF (2–3 drops) and left under a magnetic stirrer, while the development of gas was observed. After one hour, dry toluene (15 mL) was added to the solution that was evaporated under a vacuum using a rotary evaporator. The obtained orange residue was dissolved in dry DCM and added dropwise at 0 °C to a solution made of triethylamine (1.86 gm, 18.36 mmol, 3 equiv.) and commercially available (*S*)-proline methyl ester (1.4 gm, 9.18 mmol, 1.5 equiv.) in dry DCM. The reaction mixture was left under a magnetic stirrer overnight at room temperature under an N_2_ atmosphere until TLC showed completion of the reaction. The reaction mixture was diluted by the addition of DCM (20 mL) and subsequently washed with 1 N HCl (2 × 70 mL) and brine (2 × 70 mL). The organic phase was dried over MgSO_4_ and evaporated under a vacuum using a rotary evaporator to give a yellow oil. The crude of the reaction was subsequently purified by column chromatography on silica gel (mobile phase: from EtOAc, 100, to EtOAc/MeOH, 98/2, *v*/*v*) to give intermediate 1b (0.900 g, reaction yield: 36%), obtained as a yellow oil. ^1^H NMR (400 MHz, CDCl_3_) δ: as a mix of rotamers, 7.69/7.65 (s, 1H), 6.87/6.79 (s, 1H), 4.75–4.69 (m, 1H), 4.17–4.14 (t, *J* = 6.4, 2H) 4.13–4.00 (m, 1H), 3.97/3.93 (s, 3H), 3.80/3.66 (s, 3H), 3.70/3.53 (s, 3H), 3.33–3.29 (m, 1H), 3.21–3.17 (m, 1H), 2.58–2.52 (m, 3H), 2.38–2.30 (m, 1H), 2.24–2.18 (m, 3H), 2.12–1.89 (m, 4H). ^13^C NMR (101 MHz, CDCl_3_) δ: as a mix of rotamers, 173.2/173.3, 172.6/172.4, 154.8/154.2, 148.5/148.3, 137.3/137.1, 127.4/127.2, 110.2/109.5, 108.2/108.1, 68.1, 60.6, 58.5, 56.7/56.5, 52.4/52.3, 51.5, 48.3, 46.2, 31.0, 30.3/29.6, 24.5. 

Synthesis of methyl (*S*)-4-((7-methoxy-5,11-dioxo-2,3,5,10,11,11a-hexahydro-1H-benzo[e]pyrrolo[1,2-a][1,4]diazepin-8-yl)oxy)butanoate (1c):

Intermediate 1b (900 mg, 2.12 mmol, 1 equiv.) was dissolved in EtOH in a hydrogenator vial, and a catalytic amount of Raney-Nickel, slurry in H_2_O (100 mg), was added to the solution. The vial was placed in position in a Parr hydrogenation system and reacted under the pressure of H_2_ (40 psi) until TLC showed total consumption of the starting material. The reaction mixture was subsequently filtered on a Celite path, washing with DCM. The collected organic solvent was evaporated under a vacuum using a rotary evaporator to give pure 1c. If the reaction resulted in a mixture of unclosed and closed products, the crude of the reaction was refluxed in toluene after filtration of the catalyst to give conversion to the desired product 1c (0.800 g, reaction yield: 95%), obtained as an orange oil. ^1^H NMR (400 MHz, CDCl_3_) δ: 8.91 (s, 1H), 7.42 (s, 1H), 6.52 (s, 1H), 4.04 (td, *J* = 3.08, 5.92 Hz, 3H), 3.87 (s, 3H), 3.71–3.78 (m, 1H), 3.67 (s, 3H), 3.55–3.62 (m, 1H), 2.70–2.76 (m, 1H), 2.53 (t, *J* = 7.05 Hz, 2H), 2.15–2.09 (m, 2H), 1.96–2.02 (m, 3H). ^13^C NMR (101 MHz, CDCl_3_) δ: 173.5, 171.2, 165.3, 151.5, 146.6, 129.8, 119.3, 112.3, 105.1, 67.8, 56.9, 56.2, 51.7, 47.3, 30.2, 26.5, 24.1, 23.6.

Synthesis of (*S*)-4-((7-methoxy-5,11-dioxo-2,3,5,10,11,11a-hexahydro-1H benzo[e]pyrrolo[1,2-a][1,4]diazepin-8-yl)oxy)butanoic acid (1d):

Intermediate 1c (790 mg, 2.18 mmol, 1 equiv.) was dissolved in methanol (60 mL), and aqueous 1M NaOH solution (excess) was added to the reaction mixture that was stirred overnight at room temperature until TLC showed total consumption of the starting material. A rotary evaporator was used to evaporate the solvent. The obtained residue was dissolved in H_2_O (30 mL), and citric acid 1M aqueous solution was added to the solution until pH 3 was reached. The aqueous phase was extracted with EtOAc (2 × 50 mL) and the combined organic phases were evaporated under a vacuum using a rotary evaporator, affording pure 1d (0.700 g, reaction yield: 92%), obtained as a white solid. ^1^H NMR (400 MHz, MeOH-*d*_4_) δ: 7.27 (s, 1H), 6.57 (s, 1H), 4.10–4.05 (m, 1H), 3.99 (t, *J* = 6.4Hz, 2H), 3.75 (s, 3H), 3.69–3.59 (m, 1H), 3.53–3.42 (m, 1H), 3.24–3.17 (m, 2H), 2.59–2.51 (m, 1H), 2.42 (t, *J* = 7.2Hz, 2H), 2.03–1.90 (m, 4H). ^13^C NMR (101 MHz, MeOH-*d*_4_): 176.8, 172.4, 167.6, 153.5, 147.9, 132.3, 120.0, 113.1, 106.5, 69.1, 58.4, 56.6, 31.2, 27.0, 25.5, 24.6. 

Synthesis of dilactam derivatives 7a and 8a:

The Boc-protected intermediates 2a and 2b, prepared accordingly to a reported literature procedure [7] (1.2 equiv.), were dissolved in MeOH (7 mL), and HCl 4M in dioxane (7 mL) was added to the solution that was left under a magnetic stirrer for 2 h until TLC showed completion of the reaction. The reaction mixture was then evaporated under a vacuum using a rotary evaporator, giving the corresponding deprotected intermediates. 

Compound 1d (52 mg, 1 equiv.) was dissolved in DMF (7 mL), and EDCI (59 mg, 2.4 equiv.) and DMAP (50 mg, 3 equiv.) were added to the solution that was left to stir in N_2_ atmosphere for 30 min. Either the deprotected compound 2a (60 mg, 0.123 mmol, 1.1 equiv.) or 2b (55 mg, 0.124 mmol, 1.1 equiv.) was added to the reaction mixture and left to stir overnight at room temperature in the N_2_ atmosphere. The reaction did not complete and was quenched by adding H_2_O (10 mL). The aqueous phase was then extracted with EtOAc (3 × 10 mL). The organic phase was sequentially washed with citric acid 0.1 M aqueous solution (10 mL), saturated NaHCO_3_ aqueous solution (10 mL), and brine (10 mL). The collected organic phase was dried over MgSO_4_ and evaporated under a vacuum using a rotary evaporator. The crude of the reaction was purified by column chromatography on silica gel (mobile phase: DCM/Acetone, 50/50, *v*/*v*) to give the final pure products.

(*S*)-4-(4-((7-methoxy-5,11-dioxo-2,3,5,10,11,11a-hexahydro-1H-benzo[e]pyrrolo[1,2-a][1,4]diazepin-8-yl)oxy)butanamido)-1-methyl-N-(2-(thiomorpholine-4-carbonyl)benzofuran-5-yl)-1H-pyrrole-2-carboxamide (7a)

Obtained 0.040 g (reaction yield: 43%) as a yellow oil. ^1^H NMR (400 MHz, Acetone-d_6_) δ 9.25 (s, 1H), 9.22 (s, 1H), 9.12 (s, 1H), 8.21 (d, *J* = 2.01 Hz, 1H), 7.67 (dd, *J* = 8.94 Hz, 2.14 Hz, 1H), 7.48 (d, *J* = 9.06 Hz, 1H), 7.31 (s, 1H), 7.27 (d, *J* = 1.01 Hz, 1H), 7.19 (d, *J* = 1.76 Hz, 1H), 6.91 (d, *J* = 2.01 Hz, 1H), 6.74 (s, 1H), 3.93–4.11 (m, 6H), 3.89 (s, 3H), 3.80 (s, 3H), 3.54–3.65 (m, 1H), 3.43–3.52 (m, 1H), 2.68–2.75 (m, 5H), 2.47 (t, *J* = 7.18 Hz, 2H), 1.99–2.04 (m, 6H). ^13^C NMR (101 MHz, MeOH-d_4_) δ 173.7, 172.3, 167.6, 162.4, 161.8, 153.5, 153.0, 150.2, 147.8, 136.0, 132.3, 128.4, 124.5, 123.3, 122.6, 121.2, 119.8, 115.6, 113.1, 113.0, 112.7, 106.5, 106.4, 74.1, 69.2, 58.4, 56.5, 44.0, 36.9, 33.5, 30.7, 29.5, 27.0, 26.3, 24.6. HRMS (ESI, *m/z*): calc. for C_36_H_39_N_6_O_8_S_1_ ([M] + H) + 715.2545 found 715.2554.

(*S*)-N-(2-(dimethylcarbamoyl)benzo[b]thiophen-5-yl)-4-(4-((7-methoxy-5,11-dioxo-2,3,5,10,11,11a-hexahydro-1H-benzo[e]pyrrolo[1,2-a][1,4]diazepin-8-yl)oxy)butanamido)-1-methyl-1H-pyrrole-2-carboxamide (8a)

Obtained 0.040 g (reaction yield: 46%) as a yellow solid. ^1^H NMR (400 MHz, CDCl_3_) δ 9.14 (s, 1H), 8.68 (s, 1H), 8.53 (s, 1H), 8.24 (d, *J* = 1.76 Hz, 1H), 7.66 (d, *J* = 8.81 Hz, 1H), 7.46 (dd, *J* = 8.69 Hz, 1.89 Hz, 1H), 7.37 (s, 2H), 7.04 (s, 1H), 6.81 (s, 1H), 6.48 (s, 1H), 3.95–4.01 (m, 1H), 3.85–3.92 (m, 1H), 3.81 (s, 3H), 3.77 (s, 3H), 3.66–3.74 (m, 1H), 3.50–3.59 (m, 1H), 3.17 (br. s., 6H), 2.68 (dd, *J* = 6.04 Hz, 4.28 Hz, 1H), 2.29–2.37 (m, 2H), 1.83–2.12 (m, 6H). ^13^C NMR (101 MHz, CDCl_3_) δ 170.6, 170.1, 165.5, 164.9, 160.2, 151.4, 146.2, 139.2, 138.1, 135.8, 135.4, 130.2, 125.7, 123.3, 122.4, 121.6, 119.8, 119.25, 115.7, 112.4, 106.7, 104.6, 68.0, 57.0, 56.1, 47.2, 36.8, 31.8, 32.6, 29.3, 26.2, 23.6. HRMS (ESI, *m/z*): calc. for C_34_H_37_N_6_O_7_S_1_ ([M] + H) + 673.2439 found 673.2447.

### 3.3. Cell Culture and MTT Assay

WI-38 cells, obtained from ATCC, were grown in EMEM or MEM media (Gibco, Maryland, USA) supplemented with fetal bovine serum (10%, *v*/*v*, Sigma Aldrich, Dorset, UK) in an incubator which was kept at 37 °C in a humidified atmosphere containing 5% CO_2_. For the MTT cell viability assay, the cells containing fresh media were incubated with the drug for 24 h, at which point, the media was removed, and the MTT reagent was added. The cells were incubated with the MTT reagent for 2 h. After careful removal of the MTT reagent, the formazan crystals were dissolved in DMSO, and finally, the absorbance of the formazan crystals was read using a plate reader (Envision Plate Reader, PerkinElmer, Beaconsfield, UK). The absorbance values were normalised with the blank before using them for the % viability determination. Each assay was repeated six times, and the result is reported as average values. 

### 3.4. Bacterial Strains

Bacterial strains were cultured in tryptic soy broth (TSB) and on TSA plates, which were maintained at 37 °C. The strains are listed in Table 1 and Table 2. Unless otherwise stated, the chemicals used to culture the bacterial strains were purchased from Sigma-Aldrich. 

### 3.5. Susceptibility Testing

The microdilution broth method was used for antimicrobial susceptibility testing [32]. All compounds were initially dissolved in DMSO prior to dilution in broth to prepare the working solution. The effect of solvent on bacterial growth was tested as part of the study, and concentrations of solvent that had no effect on bacterial growth were used for the sample preparation. The MIC was defined as the lowest concentration of compound which resulted in no visible growth at an optical density of 600 nm. The susceptibility testing was performed in triplicate, and the MIC range is provided in Table 1 and Table 2. 

### 3.6. Molecular Docking of Compounds 7 and 7a with DNA

The NAB module of the AMBER 12.0 package program was used to generate Double Strand (DS) DNAs type B (BDNA) using Seq-1: 5′-TAT-AGA-TAT-AGA-TAT-3′ and Seq-2: 5′-CGC-TAT-AGA-TAT-CGC-3′ sequences. ChemDraw 21.0 was used to prepare the ligand files for docking. SYBYL software was used to optimise the DNA and ligands prior to molecular docking. Initially, blind docking by SMINA was performed to identify the best binding site of ligands in the generated DNA. All the probable DNA binding modes were considered during the blind docking. Finally, AutoDock 4.0 was used to perform the molecular docking in the SMINA-identified binding site by keeping all parameters at the default value.

### 3.7. Molecular Docking of the Compounds to Gyrase A

Blind molecular docking of compounds 7 and 7a to the gyrase A from *S. aureus* (PDB ID 2XCT) was performed by AutoDock SMINA by keeping all parameters at their default values. This allowed us to find the best binding pocket by exploring all probable binding cavities in the enzyme. GOLD molecular docking software was used for the molecular docking of compounds 7 and 7a into the SMINA-located binding site. The best-docked pose for each compound was selected after evaluating both the fitness function score and ligand binding position.

## 4. Conclusions

Two dilactam analogues of previously reported pyrrolobenzodiazepine monomers, 7 and 8, were synthesised using a medicinal chemistry approach to evaluate the importance of covalent DNA binding on the antibacterial activity of this new broad-spectrum antibiotic class. The dilactam analogues, 7a and 8a, were found to be active against MDR Gram-positive bacteria but lost activity against the target MDR Gram-negative bacteria. This suggests covalent DNA binding is critical for the Gram-negative activity as the dilactams lack the N10-C11 imine, which is responsible for covalent interaction with the guanine base of DNA. The moderate retention of activity against Gram-positive bacteria is encouraging and offers a chemical scaffold for further medicinal chemistry optimisation to develop antibiotics that target Gram-positive bacteria.

## Data Availability

All data related to this study can be found in the main manuscript and the Appendix A.

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
