# Peer review of "Covalent DNA Binding Is Essential for Gram-Negative Antibacterial Activity of Broad Spectrum Pyrrolobenzodiazepines"

_antibiotics, 2022, doi:10.3390/antibiotics11121770_

Round 1
Reviewer 1 Report
The work presented by the authors is interesting material, although it covers the study of only two molecules, however, it represents the value for the medical chemistry of antimicrobial agents and fits to the profile of the journal. I would recommend this manuscript to be published, but major revision should be done following the comments below:
Abstract:
1. Rewrite the first sentence of the abstract – it’s very similar to the beginning of the abstract in the article [7] and, honestly, the problem emphasized in this sentence is not solved by current work.
Introduction section:
1. Please, use the abbreviation “MDR” (as well as other abbreviations) everywhere suits instead full-words versions, or don’t follow using abbreviations at all.
2. Please clearly state in this part of the paper which synthetic strategies have been carried out and reported in the literature to modify the imino-moiety of this class of compounds in order to the structure optimization and what effect this has had on the antimicrobial activity
3. The last two sentences of the "Introduction" section are not appropriate for this part of the paper, they are more appropriate for the discussion or conclusion section. Therefore, it is worth removing them from here.
4. Figure 1. Please indicate the stereochemistry of ”Compound 8” like in ”Compound 7”.
5. I suggest changing the numbering of lead compounds 7 and 8 (Fig.1 and below through the text). The use of numbers 7 and 8, in my opinion, introduces confusion in the perception of work. Maybe there are some laboratory code names for these molecules?
Results and Discussion
1. Scheme 1. Please, renumber compounds 1a-d on 1, 2, 3, and 4. Use of "1a-x" style fits for a presentation of a series of structure–related derivatives.
2. Scheme 2. Please, change the scheme. Should be presented two individual reactions lead to the obtaining of two individual molecules.
3. Scheme 2. In the description of the scheme, there is an incorrect numbering of derivatives.
4. I suggest changing the numbering of new compounds 7a and 8a, in my opinion, it introduces confusion in the perception of work.
5. Table 1. and Table 2. – Data presented for four molecules, so, please, note all of them in the title.
6. Figure 2. Represent the data in the figure so that the leftmost column is for the molecule with a smaller number and follow to a larger number from left to right.
7. I suggest to the authors add the “Conclusions” section (3-4 sentences) to the present draft. The work only benefits from this!
Materials and Methods
1. Please, create and format subchapters to describe all synthetic and biological research conducted according to the publisher's requirements all.
2. Please, in the description of 13C NMR spectra use “CDCl3” instead “CHLOROFORM”.
There are technical typos in the text in the form of extra spaces and dots, please clean the text from them!
Author Response
Please see the response in the attached file.

Reviewer 2 Report
This is a straightforward manuscript, reporting the synthesis and antibacterial evaluation of two pyrrolobenzodiazepine-diones. Against a panel of Gram-positibe bacateria, the compounds were found to have lower in vitro activity than the parent pyrrolobenzodiazepine-ones, and against selected Gram-negative bacteria, the new compounds were found to be inactive. I recommend publication the journal Antibiotics, after the points listed below have been taken into account.
- The title reads like a conclusion and is confusing, as it indicates that the compounds containing a N10-C11 iminine group were studied in the present work, which was obviously not the case. I suggest something like: "Synthesis and in vitro antibacterial evaluation of pyrrolobenzo-1,4-diazepin-diones" or so.
- "an MIC" should be "a MIC" throught. For the sake of clarity "MIC" should be "MIC90" or whatever has been determined.
- Page 2: Are you sure that ref. 21 proves that PBD-dimers cannot permeate Gram-negative membranes (i.e. cell wall?)
- Numbering of the compounds studied is strange. It is odd that they start with 7 and 8 in Figure 1. Moreover, there are just 1,2, 7 and 8, whereas 3, 4, 5, and 6 do not occur. Consider revising.
- According to textbook conventions, 36 % or 43-46 % can hardly be called "good yield": https://en.wikipedia.org/wiki/Yield_(chemistry)#Definitions
- Page 3, second paragraph: The last sentence needs a citation. Moreover, "according to procedure" should be "according to a procedure".
- Caption to Scheme 2: The meaning of "4.15-4.16" is not clear.
- Page 4, second paragraph, last sentence: I would be a little more reluctant. Maybe "suggests" or "should indicate" rather than "clearly indicates". As far es I see, there is no definite proof for the statement.
- Page 6: A subsection "Physical methods" or so should be separated from the "Genertal chemistry".
- Page 6,last paragraph: "obtained accordingly literature procedure" should be "obtained according to a literature procedure"and a citation is needed.
- Page 6 "exracted with 1 N HCl" should be "washed with 1 N HCl".
- "CHLOROFORM-d" should be "chloroform-d" or "CDCl3" throughout.
- Page 8, second paragraph: "effect on solvent on" should be "effect of solvent on".
- Page 8: "Molecular docking" should have a separate subsection. Why not dock also compounds 8 and 8a, using the same procedure?
- Supporting Information: 1H and 13C NMR spectra and HPLC chromatograms should be depicted.
Author Response
Please see the point by point response in the attached file.

Reviewer 3 Report
The manuscript send by Picconi et al. describes the design and synthesis of two dilactam analogues of pyrrolobenzodiazepine (PBD), that were tested for their antibacterial activity. These dilactam analogs do not bear an imine moiety unlike their PBD analogs, which aim to prevent cytotoxicity. The MICs of these analogs were evaluated on 6 Gram-positive strains and 6 Gram-negative strains and compared to their parent PDBs. The dilactam analogs showed reduced antibacterial activities compared to the PDBs with moderate MIC on Gram-positive strains and inactivity on Gram-negative strains. This is due that the imine cannot covalently bind to DNA. Indeed, this is not surprising due to the mode of action of PDBs, which has already been describes as covalently interacting via this imine (cf. Antonow et al. Biochemistry 2008, 47, 11818–11829 and references herein (this article should be cited)). A molecular modelling study was performed and revealed that the amide instead of the imine could also affect the interaction of the compound with DNA gyrase. This imine is therefore crucial for broad-spectrum activity but may be interesting for targeting Gram-positive strains as these compounds are not cytotoxic.
The conclusion of this article is interesting even though the importance of the imine is quite obvious from the mode of action of this chemical series as they are know to be covalent inhibitors via this moiety.
The work is clearly presented and the conclusions are reasonably supported by the data.
Two points needs to be addressed:
- The experimental procedures need to be revised to add characterization of the new compounds. There should be LCMS, 1H and 13C NMR spectra, but also either HRMS or elemental analysis for the final compounds. In addition, the melting point for solids is also advised.
- The self-citation rate is quite high and could be decreased.
Please find below some comments:
Introduction
· The year of the discovery of the daptamycin should be added in order to estimate more easily the period of discovery void.
· For a better understanding, a figure could be added or the figure 1 could be modified in order to explicit the numbering of the atoms of the PDB ring and to represent the position of the 3rd aliphatic ring.
· A reference should be added for the toxicity associated with PDBs.
· “Compound 7 and 8 (Figure 1) which were identified as lead compounds with excellent activity against both Gram-positive and Gram-negative bacteria.” Please add a reference.
Results and discussion
Synthesis:
· “The synthesis of dilactam analogues of compounds 7 and 8 was based”: please replace 7 and 8 by 7a and 8a.
· The authors need to qualify their comments on the yield of the first step (36%) or explain why this yield is considered good.
· Please replace “reduction of the nitro group of the derivative 1c” by “reduction of the nitro group of the derivative 1b”
· The authors should clarify the cases where the heating is needed to synthesize the compound 1c. Is is a scale-up problem, a catalyst problem and/or for other PDb compounds?
· Revise the title of Scheme 2 (4-15-4.16 is written instead of 7a and 8a)
· The authors need to qualify their comments on the yield of the coupling (43%-46%) or explain why this yield is considered good.
· The paragraph concerning the free-protecting group procedure should be put in perspective of other cases in order to help the reader if it is uncommon or not for PDBs synthesis.
· These sentences can be removed as they afford obvious information: “The intermediate of the synthesis of the core and the final products were fully characterized by LC-MS, HRMS and 1H and 13C NMR analysis (ESI). Compounds 7 and 8 were synthesized according to procedure recently described by us.”
MIC results:
· The choice of the different strains could be discussed.
· “between 4 (or 8) and 32 μg/mL”: please replace by between 4 (or 8) and more than 32 μg/mL
· Please revise the legends of Table 1 and Table 2 to mention also the compounds 7 and 8.
· Please add an abbreviation section in order to explicit the different strains.
Molecular modeling
· Please replace “DNA Gyrase enzyme” by “DNA gyrase enzyme”
· For a better understanding, please explain why Ser85 is a critical amino acid.
Materials and Methods
· Synthetic procedures and analytical characterization of compounds 2a, 2b, 7a and 7b are missing. Please add this crucial information.
· Please add for all the procedures the real used quantity (mass and mol) of compounds instead of equivalent
· Please add LCMS analysis for all compounds
Synthesis of 1b
· Please replace “MgSO4” by “MgSO4”
· The letter size of the NMR description is higher than the other. Please rectify.
· Please add the coupling unit (Hz) for the 4.14 ppm peak.
· The 1H and 13C NMR are not clear. There are more described protons and carbons than there are on the molecule.
Synthesis of 1c
· Please replace “Intermediate 1c” of the first line by “Intermediate 1b”.
Synthesis of 1d
· 1C is missing
Cell culture
Please replace “were grown EMEM” by “were grown in EMEM”
References
Volume, Issue and pages are missing for reference 7
Last page is missing for reference 10
Self-citation is quite high (11 references on 25 = 44%). Most recent articles of the authors should be kept and some contributions from other groups could be also cited such as:
Anticancer and ADC:
Hartley JA. Expert Opin Investig Drugs. 2011 Jun;20(6):733-44
Gregson SJ, Masterson LA, Wei B, Pillow TH, Spencer SD, Kang GD, Yu SF, Raab H, Lau J, Li G, Lewis Phillips GD, Gunzner-Toste J, Safina BS, Ohri R, Darwish M, Kozak KR, Dela Cruz-Chuh J, Polson A, Flygare JA, Howard PW.J Med Chem. 2017 Dec 14;60(23):9490-9507.
Cipolla L, Araújo AC, Airoldi C, Bini D. Anticancer Agents Med Chem. 2009 Jan;9(1):1-31.
Donnell AF, Zhang Y, Stang EM, Wei DD, Tebben AJ, Perez HL, Schroeder GM, Pan C, Rao C, Borzilleri RM, Vite GD, Gangwar S.Bioorg Med Chem Lett. 2017 Dec 1;27(23):5267-5271.
Jackson PJM, Kay S, Pysz I, Thurston DE.Drug Discov Today Technol. 2018 Dec;30:71-83.
Li Y, Quan J, Song H, Li D, Ma E, Wang Y, Ma C.Bioorg Chem. 2021 Sep;114:105081.
Antibacterial:
Iacobino A, Giannoni F, Fattorini L, Brucoli F.J Antibiot (Tokyo). 2018 Sep;71(9):831-834.
Author Response

(The authors gave the same response as above.)

Round 2
Reviewer 1 Report
The authors made some changes in the draft but it could not be considered a “significant improvement”.
The authors did not make relevant corrections to the draft in accordance with the following comments:
Introduction section:
2. Please clearly state in this part of the paper which synthetic strategies have been carried out and reported in the literature to modify the imino-moiety of this class of compounds in order to the structure optimization and what effect this has had on the antimicrobial activity
5. I suggest to change the numbering of lead compounds 7 and 8 (Fig.1 and below through the text). The use of numbers 7 and 8, in my opinion, introduces confusion in the perception of work. Maybe there are some laboratory code names for these molecules?
Results and Discussion
1. Scheme 1. Please, renumber compounds 1a-d on 1, 2, 3, and 4. Use of "1a-x" style fits for a presentation of a series of structure–related derivatives.
2. Scheme 2. Please, change the scheme. Should be presented two individual reactions lead to the obtaining of two individual molecules.
3. I suggest changing the numbering of new compounds 7a and 8a, in my opinion, it introduces confusion in the perception of work.
6. Figure 2. Represent the data in the figure so that the leftmost column is for the molecule with a smaller number and follow to a larger number from left to right.
Materials and Methods
2. Please, in the description of 13C NMR spectra use “CDCl3” instead “CHLOROFORM-d”. – Authors indicated that they corrected this, however, it is not true!
Additionally, please put in superscript “1” and “13” in the description of all 1H and 13C NMR spectra.
Conclusions
1. The conclusions provided by the authors are extremely difficult for understanding! First of all due to the use of irrational numbering of described compounds!
2. Else one question for the “Conclusions” section: what do authors mean by the term: “traditional medicinal chemistry approach”?
3. Conclusions should be improved!
Taking into the account all above mentioned, the above shortcomings are not acceptable for high-level articles and the manuscript in its current form cannot be recommended for publication in a journal of such rank as Antibiotics. Therefore, I insist on correcting all the comments made at the previous stage and added at this stage of the review, and major revision of the draft should be performed again!
Reviewer 3 Report
I would like to thank the authors to have consider most of the previous comments.
In order to perfect the manuscript, please find some suggested minor corrections to be made.
Introduction :
Please remove the dot between “4” and “in” in : “daptomycin4. in the late 1980s”
Figure 1: Please replace “pyrrobenzodiazepines” by “pyrrolobenzodiazepines”
Results and discussion:
Synthesis
Please put a dot at the end of the sentence “…where Alloc- THP protection is required “
MIC results
Concerning the compound 7a. Please replace “Gram-positive strains, with MIC values between 4 and 32 μg/mL” by “Gram-positive strains, with MIC values between 4 and 32 μg/mL”
Material and methods
Synthesis 7a and 8a
Please replace “boc” by “Boc”
Please add mass and molar information about 2a and 2b
Please replace “Dioxane” by “dioxane”
Please add molar information for reactants and reagents for homogeneity consideration
Please replace “AcOEt” by “EtOAc”
Please replace “Citric Acid” by “citric acid”
Please replace “Brine” by “brine”
Please replace “1H” by “1H”
Please replace “d6” by “d6”
Please replace “CHLOROFORM-d” by “CDCl3”
Author Response
Please see the attached response.

Round 3
Reviewer 1 Report
The authors of this manuscript do not show a significant willingness to improve the draft and in the proposed/submitted shape the manuscript can’t be published in journal “Antibiotics”.
My decision is “to reject” this manuscript.